# A SINGLE CHARACTER CAN MAKE OR BREAK YOUR LLM EVALS

## ABSTRACT

Common Large Language model (LLM) evaluations rely on demonstration examples to steer models' responses to the desired style. While the number of examples used has been studied and standardized, the choice of how to format examples is less investigated. In evaluation protocols and real world usage, users face the choice how to separate in-context examples: use a comma? new line? semi-colon? #, | etc? Surprisingly, we find this seemingly minor choice can dramatically alter model response quality. Across leading model families (Llama, Qwen, Gemma), performance on MMLU for example can vary by $\pm 23\%$ depending on the choice of delimiter. In fact, *one can manipulate model rankings to put any model in the lead* by only modifying the single character separating examples. We find LLMs' brittleness pervades topics, model families, and doesn't improve with scale. By probing attention head scores, we find that good-performing delimiters steer attention towards key tokens in the input. Finally, we explore methods to improve LLMs' robustness to the choice of delimiter. We find specifying the selected delimiter in the prompt boosts robustness and offer practical recommendations for the best-performing delimiters to select.

## 1 INTRODUCTION

Prompting a language model is a fickle craft. The quality of a language model's response is sensitive to how a user crafts their prompt (Liu et al., 2023). Commonly reported large language model (LLM) evaluations, however, use fixed prompt templates that do not capture models' sensitivity to prompts in real world usage. This blind spot in LLM evaluations affects how we measure progress, compare models, and research new ways of training models. To muddy matters further, LLM evaluation protocols for many benchmarks rely on demonstration examples to steer model outputs. The intent is to mirror how real world users provide a few examples of the response type, style, or desired format.

Numerous studies have explored prompt engineering choices related to demonstration examples, such as incorporating chain-of-thought (Wei et al., 2022) instructions with demonstration examples and standardizing the number of demonstration examples in evaluation protocols (Brown et al., 2020; Min et al., 2022; Agarwal et al., 2024). One understudied choice, however, is that of the delimiter used to separate demonstration examples. In evaluation protocols and real world usage, users face the choice of how to separate examples: use a comma? new line? semi-colon? or one of the many other characters "#", "|", etc.? Surprisingly, we find this seemingly minor choice of the single character separating examples can render common LLM evaluations unintelligible.

We first establish a common evaluation protocol to offer a clear basis for comparisons across and within models. Wl unify evaluation protocols using the popular open-sourced codebase powering the Open-LLM-leaderboard.[1] With other choices fixed, we turn to studying the effect of the single character separating demonstration examples, which we call *example delimiter* (shown in Figure 1). This study does not modify the content of the question or examples, but only changes the character used to separate examples. We choose a set of representative benchmarks in our evaluation, including MMLU (Hendrycks et al., 2021), ARC-challenge (Clark et al., 2019), and commonsense-QA (Talmor et al., 2019), following Zheng et al. (2024). Then we systematically evaluate the effect of 30

---

[1] https://huggingface.co/spaces/open-llm-leaderboard/open_llm_leaderboard

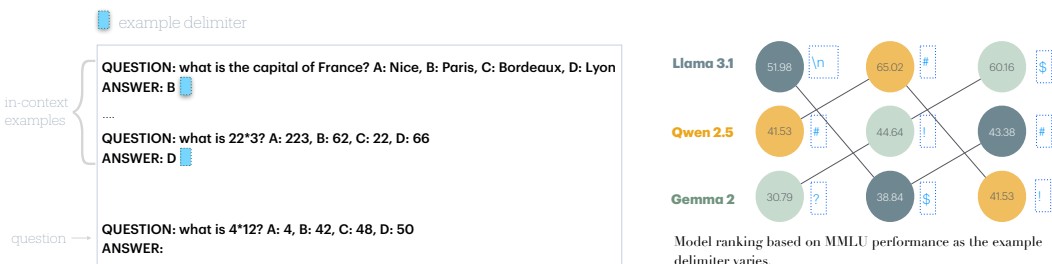

Figure 1: **One can manipulate rankings to put any model in the lead by varying the single delimiter character.** On the left, we show the delimiter used to separate examples in common evals with few-shot examples such as MMLU. On the right, we show model rankings based on MMLU performance as the example delimiter varies with each column corresponding to a different ranking.

non-alphanumeric ASCII *example delimiters* on leading models, including Llama, Gemma, and Qwen.

We find that a single *example delimiter* can dramatically alter model response quality. For example, changing a single *example delimiter* leads to $18.3\% \sim 29.4\%$ MMLU performance differences on all leading models studied in this work (2). This performance gap is equivalent to three years of cumulative progress in language models since April 2022 [2]. We find similarly large fluctuations across other benchmarks when we vary the *example delimiter*, including MMLU, ARC-challenge, commonsense-QA, in-context few-shot classification (Casanueva et al., 2020; Zhang et al., 2017) and dictionary lookup tasks (Chen et al., 2024). In fact, we find that *one can manipulate the rankings to put any model in the lead* by only modifying the single-character example delimiter.

Next, we explore methods to boost LLMs' robustness to the choice of delimiter. With a proper *example delimiter* in mind, we find that specifying this *example delimiter* in the prompt consistently boosts the robustness. For example, Qwen2.5-7B-instruct gains +14.2% on MMLU with this simple prompt modification. Without specifying the example delimiters, we find that "\n" and "!" are excellent choices that recover a good overall performance.

Beyond these immediate practical recommendations, to better understand the mechanism driving model performance, we study how attention heads respond to the choice of delimiter. Specifically, we probe attention head scores on the dictionary search task where demonstration examples are crucial. We find that the right choice of delimiter can steer models' attention to focus on the relevant parts of demonstration examples — with a statistically significant gain in attention scores for key tokens needed to solve the task.

LLMs' sensitivity to the choice of delimiter rests on a complex interaction among queries, training data, model architecture that warrants further research. The example delimiter study in this work reveals that we have not fully understood the learning dynamics of modern language models.

## 2 RELATED WORK

### 2.1 PROMPT SENSITIVITY

LLM prompt sensitivity refers to how changes in the input prompt—especially minor or seemingly irrelevant ones—can lead to significant differences in a model's output. This includes variations in formatting (e.g., bullet points vs. paragraphs, delimiters, punctuation or casing), in the ordering of the few-shot examples, in wording (e.g., "Tell me" vs. "Please describe"), and in the presence or absence of specialized tokens or instructions.

---

[2]According to PapersWithCode (https://paperswithcode.com/sota/multi-task-language-understanding-on-mmlu), see Figure 7 in the appendix.

**Prompt formatting** A growing body of work shows that LLMs are acutely sensitive to superficial aspects of prompt formatting, raising concerns about the robustness and interpretability of benchmarks when using in-context learning. Zhao et al. (2021) demonstrated that small changes in prompt structure—such as label choices or phrasing—can lead to performance shifts of up to 30%, which can be partially mitigated through calibration. Sclar et al. (2024) shows prompt formatting can lead to variation of up to 76 points in LLM performance on a set of tasks from SuperNatural-Instructions (Mishra et al., 2021). Building on this, Lu et al. (2022) systematically studied the effect of example order in few-shot prompting, showing that permutations alone can cause large accuracy swings and that optimal orders often do not transfer across models. In contrast, Bertsch et al. (2024) found that long-context models—when given hundreds of demonstrations—become substantially less sensitive to such order effects, suggesting that scaling context length can mitigate formatting brittleness. Min et al. (2022) extended this by finding that models benefit from demonstrations even when they are semantically irrelevant, suggesting that formatting cues play a more central role than content alignment. Webson and Pavlick (2022) further challenged the assumption of semantic understanding, showing that models can maintain high performance even when the prompt instructions are nonsensical, indicating reliance on shallow statistical patterns. Finally, Brown et al. (2020) established the paradigm of few-shot prompting itself by manually formatting input–output exemplars within prompts, highlighting that the very act of structuring examples inline is central to enabling in-context learning. These results collectively underscore how our own observation—that merely changing the separator between few-shot examples can shift accuracy by 30 points—fits into a wider pattern of extreme sensitivity to formatting.

**Prompt strategies** While work described in the previous paragraph primarily focuses on optimizing few-shot prompting, minor changes in wording have also been shown to significantly influence performance. Wei et al. (2023) demonstrated that inserting brief phrases such as "let's think step by step," can unlock markedly better performance on various reasoning tasks, suggesting that minimal prompt edits can activate latent capabilities in large models, a technique now known as chain-of-thought (CoT) prompting. However, Turpin et al. (2023) showed that CoT outputs are often unfaithful to the model's true reasoning process and are sensitive to small perturbations in phrasing, highlighting the fragility of generated rationales. Elhage et al. (2022) further provided a mechanistic account of how trivial input patterns can become disproportionately influential during training, especially during transitions like grokking, when models shift from memorization to generalization. Zhao et al. (2023) highlight that carefully crafted bias-inducing instructions can systematically steer model outputs, underscoring that prompt strategies can encode not only reasoning scaffolds but also social and ethical biases. This brittleness can be partially mitigated through strategies such as Self-Consistency prompting, which improves reasoning reliability by sampling multiple diverse chains of thought and aggregating their final answers via majority voting, reducing variance and stabilizing outputs (Wang et al., 2023). Scratchpad prompting encourages models to generate structured intermediate computations during inference, enabling more accurate multi-step problem solving by externalizing latent reasoning steps (Nye et al., 2021). Beyond strategies that are supposed to improve reasoning, there are several context dependent prompting strategies:

- **Instruction prompting** improves model alignment and generalization by training or prompting models with natural language task descriptions, as demonstrated by T5, FLAN, and related systems (Sanh et al., 2022). Reformulation strategies such as ReAct interleave reasoning and action steps, enabling models to reason while interacting with tools or environments (Yao et al., 2023b).

- **Meta prompting** approaches like Tree-of-Thoughts structure reasoning as an explicit search over multiple intermediate states, allowing backtracking and pruning of poor reasoning paths (Yao et al., 2023a). Program-aided prompting similarly integrates symbolic computation into the reasoning process, enabling stronger performance on algorithmic or math-intensive tasks (Gao et al., 2023).

- **Persona-based prompting** conditions the model's output by assigning it a specific role or identity (e.g., expert, tutor), which can improve coherence, calibration, and task relevance (Zhou et al., 2019). Debate and Socratic prompting elicit richer and more nuanced responses by framing generation as a multi-perspective dialogue or question-driven exploration (Lazaridou et al., 2022).

- **Gradient-based prompt search** methods like AutoPrompt (Shin et al., 2020) learn discrete prompts by optimizing input token embeddings directly for task performance. In contrast, soft prompt tuning and prefix tuning prepend learned embeddings to the input sequence, conditioning model behavior without updating the full model. Reynolds and McDonell (2021) further illustrate that even manual prompt crafting—choosing phrasing and structure heuristically—can deliver strong results, motivating the study of automated search methods.

- **Deliberation prompts** encourage models to critique, revise, or self-evaluate their initial outputs (Madaan et al., 2023), often improving factual accuracy and reasoning quality. Confidence-aware prompting strategies ask models to estimate uncertainty or express confidence in their answers, improving interpretability and enabling more robust downstream decision-making.

Practitioners have also collated prompt strategy guides including Anthropic, OpenAI, and Boonstra (2025) which describe heuristics for how to encode context, formulate questions, and include instructions for better model performance.

## 2.2 BRITTLENESS OF BENCHMARKS

The leaderboard illusion shows model ranking brittleness when controlling for access to data and proprietary evaluation protocols (Singh et al., 2025). In a similar vein, Chandak et al. (2025) re-evaluate the performance gains coming post-reinforcement learning training in the context of mathematical benchmarks for LLMs. Prior work quantified the effect of input perturbations on language model performance showing model rankings are sensitive to prompt format perturbations (Sclar et al., 2023). Complementary to these findings, Bowyer et al. (2025) caution that standard statistical tools such as the Central Limit Theorem can yield misleading uncertainty estimates when benchmarks contain only a few hundred datapoints, further undermining the reliability of leaderboard comparisons. Beyond formatting perturbations and statistical fragility, recent work has underscored that benchmark design itself can be inherently brittle. Hardt (2025) argues that static test sets, while historically instrumental, often incentivize overfitting, exploitation of dataset artifacts, and prioritize leaderboard performance over genuine generalization—reflecting Goodhart's Law. He further notes that while model rankings were surprisingly stable across datasets in the ImageNet era, modern multitask LLM benchmarks exhibit marked instability (e.g., adding weak models can reorder leaderboards), and the danger of test-set memorization—especially under massive pretraining—further undermines evaluation validity.

## 3 METHOD: A COMMON EVALUATION PROTOCOL REFLECTING REAL WORLD USAGE OF LLMS

In the prompt of a language model, demonstration examples can either help provide new knowledge (e.g. few-shot learning examples of a new task (Casanueva et al., 2020)) or establish the expected response style (e.g. a question followed by four potential answers 'A'-'D' and one final answer (Hendrycks et al., 2021)). On the one hand, providing such examples in the prompt has been shown to be effective across various benchmarks (Chen et al., 2024). On the other hand, real-world prompts might also provide some examples for the same purpose. Consequently, one needs to decide how to separate these demonstration examples from one another — using a comma? new line? semi-colon? or "|", "#", etc.?

Both real-world users and language model developers hope that the models are robust to such minor formatting changes. In this work, we carefully study whether this is in fact the case. We construct a realistic yet simple evaluation pipeline, choose a set of representative benchmarks, and evaluate on a diverse set of instruction-tuned language models. The three components are as follows:

**Language models**   Compared with base language models, instruction-tuning enhances language models' ability to generate plausible and human-like outputs, attracting significant attention. Therefore, we choose a diverse set of instruction-tuned open-source language models from the Llama, Gemma, and Qwen families. Specifically, in this work, we consider two model sizes, approximately 8B and 70B. The smaller size (Llama-3.1-8B, Gemma-2-9B, and Qwen2.5-7B) represents models runnable on a modern laptop, while the larger size (Llama-3.1-70B) helps us assess the effect of scaling model sizes on our findings. While we do provide some results using GPT-4o in Appendix

Table 1 and Table 37, we focus on open-source models due to their transparency in text generation over closed-source alternatives (e.g. GPT). For example, closed models such as those behind OpenAI APIs (OpenAI, 2025) incorporate additional undisclosed pre-processing steps and a routing procedure across multiple models, which introduces confounding factors to our scientific findings.

**Representative benchmarks** We select widely used benchmarks, including MMLU (Hendrycks et al., 2021), ARC-challenge (Clark et al., 2019) and commonsense-QA (Talmor et al., 2019), to assess language model performance under different demonstration separators. Surprisingly, we find that this seemingly trivial choice leads to significantly different performance on leading benchmarks. To further explore the underlying reasons for the single separator's impact on language model performance, we incorporate classic multi-class classification in-context (Casanueva et al., 2020; Zhang et al., 2017) and the dictionary lookup task (Chen et al., 2024).

**A realistic evaluation pipeline** During the evaluation of MMLU and ARC-challenge and commonsense-QA tasks, we incorporate a standardized evaluation pipeline. That is, for all instruction-tuned models used in this work, we use the chat template by directly appending task-specific demonstration examples (e.g., "1+1=? A: 1, B: 2, ...") together with the question as the user-role message, mirroring how real-world users provide examples before asking a question. Then, we feed this prompt to the model and evaluate the corresponding outputs. Compared with other evaluation pipelines (Wu et al., 2025; Yun et al., 2025; llama cookbook), this naive evaluation pipeline does not aim to achieve the best scores for any model, but instead aims to provide a consistent protocol for comparison and align with real-world usage.

# 4 EXPERIMENTS: CHANGING A SINGLE DELIMITER CHARACTER CAN DRAMATICALLY CHANGE PERFORMANCE ON LEADING BENCHMARKS

To assess the effect of different example delimiters, we evaluate 30 non-alphanumeric characters, including question marks, exclamation marks, commas, hashtags and so on, as listed in Table 4 in the Appendix. We then report the performance spread (max - min) across all these choices. Of course, enriching the example delimiter set, e.g., HTML-like tags, can increase the performance gap between the best and worst delimiter. However, this small non-alphanumeric delimiter set has already showcased a huge performance gap.

**Delimiters affect model outputs across benchmarks** Figure 2 shows the performance of Llama-3.1-8B-instruct, Qwen2.5-7B-instruct, and Gemma-2-9B-instruct on common benchmarks (MMLU, ARC-challenge, commonsense-QA), where the performance fluctuates considerably depending on the choice of delimiter. Specifically, we observe MMLU performance drops 18.3% for Llama-3.1-8B-instruct, 23.5% for Qwen2.5-7B-instruct, and 29.4% for Gemma-2-9B-instruct. More importantly, many semantically meaningful delimiters also suffer from this fluctuating behavior, including "&" which is commonly used to separate a list of items, and "#" which is commonly used in the markdown format. A similar fluctuation also exists in ARC-challenge and commonsense-QA tasks.

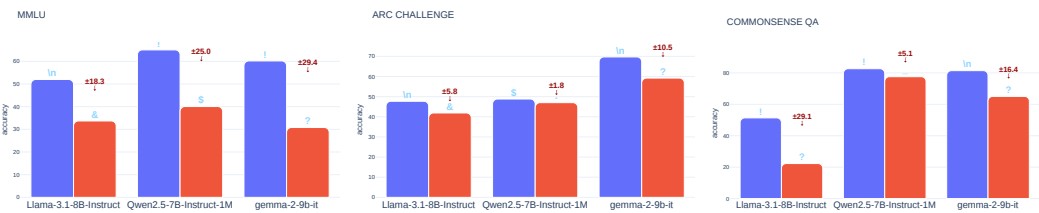

Figure 2: **Changing a single delimiter character can dramatically change performance across model families.** We show model performance across Llama, Qwen, and Gemma families on MMLU, ARC-challenge, and commonsense-QA as we vary only the example delimiter (shown above each bar in blue).

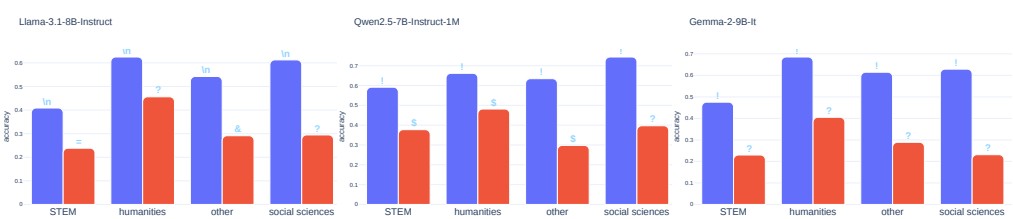

Figure 3: **The choice of delimiter affects performance across a range of topics.** We show the accuracy by topic for MMLU across three model families. The choice of delimiter (shown above each bar in blue) affects performance across a range of topics across the three model families.

**Delimiters affect model outputs across topics**    Figure 3 provides a study of the effect of delimiters across various topics within MMLU, ranging from history and philosophy to science and math. We find these fluctuations are widespread across the range of topic domains in MMLU, suggesting the sensitivity is widespread across topics as shown in Figure 3.

In fact, as shown in the right panel of Figure 1, we find that *one can manipulate model rankings to place any model in the lead* only by modifying the choice of delimiter.

### 4.1 LLMs' BRITTLENESS TO THE CHOICE OF DELIMITER IS PERVASIVE

**Scaling Llama from 8B to 70B does not improve brittleness to the choice of delimiter**    As shown in Figure 4, we compare the performance spread of Llama-3.1-8B-instruct with that of Llama-3.1-70B-instruct, a model nearly $9\times$ larger. Although the larger model achieves better performance for all three benchmarks, it doesn't increase the robustness on the choice of delimiters. For instance, on commonsense-QA task, a larger Llama-3.1-70B-instruct model shows a more serious brittleness than a smaller Llama-3.1-8B-instruct model (Figure 4 right plot, $\pm40\%$ vs $\pm29.1\%$). This finding suggests that, *while scale can boost overall performance, scale alone does not address the model brittleness to the choice of delimiter.*

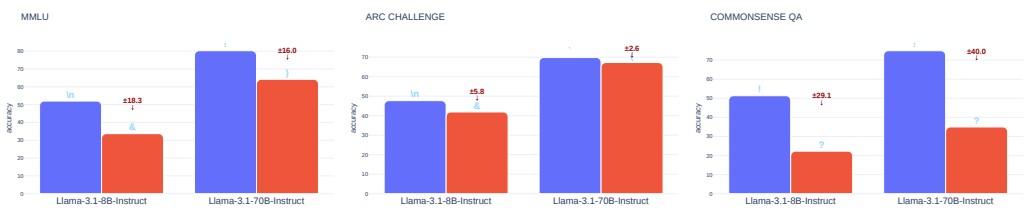

Figure 4: **Larger models are just as brittle to the change in delimiter.** We compare the performance of Llama-3.1-instruct across two sizes 8B and 70B as the delimiter varies (shown above each bar in blue). We find model scale despite improving overall performance across all three benchmarks, the larger Llama model is just as susceptible to the choice of delimiter, with a fluctuation on commonsense-QA of 40% (an even larger change compared to the smaller model).

**Closed GPT-4o also exhibits brittleness to the choice of delimiter**    Although our study focuses primarily on open-source models, one may ask whether the delimiter brittleness study extends to closed-source models. We answer this affirmatively by presenting the MMLU accuracy of GPT-4o across the same sets of delimiters in Table 1. Remarkably, GPT-4o demonstrates a spread of 45.63%, which is nearly $3\times$ higher than that of other open-source models we have considered. This evidence suggests that *delimiter brittleness is not limited to open-source models, but is an issue that persists across both open- and closed-source models.*

**Models remain sensitive to the choice of delimiter even as the number of demonstration examples increases.**    While the number of demonstration examples for MMLU, ARC-challenge, and

Table 1: MMLU summary statistics under different delimiters of GPT-4o.

| Model | Highest | Lowest | Mean $\pm$ Std | Spread |
|---|---|---|---|---|
| GPT-4o | 78.59 ("@") | 32.97 ("?") | $69.30 \pm 8.80$ | 45.63 |

commonsense-QA have commonly adopted standards that we use throughout our experiments, other classic in-context learning classification tasks (Casanueva et al., 2020; Zhang et al., 2017) can be studied with a variable number of demonstration examples. Here we explore whether the same brittleness to delimiters holds as we vary the number of examples for classic in-context learning tasks. Following Li et al. (2024) and Zhang and Bottou (2025), we choose two multi-class classification tasks: Banking77 (Casanueva et al., 2020) and Tacred (Zhang et al., 2017). Then we replace the original semantic target labels (e.g. "happy", "angry") with anonymous target labels (i.e. "class_00", "class_01") to directly access in-context learning ability. We measure the performance as we vary the number of demonstration examples from two to ten. We find, as shown in Figure 5, performance for in-context learning tasks can also vary dramatically depending on the choice of delimiter. For example, on Banking77, the performance of Llama-3.1-8B-instruct can vary between $20\%$ and $80\%$ depending on whether "[space]" or "\n" is selected as the delimiter.

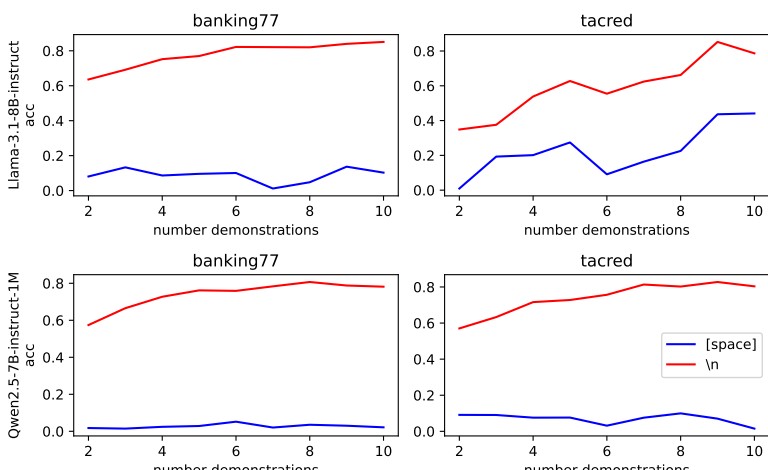

Figure 5: **The effect of delimiter ("[space]" or "\n") on Llama-3.1-8B-instruct and Qwen2.5-7B-instruct in-context learning performance.** Delimiter dramatically changes the in-context learning performance regardless of model or the number of demonstrations.

## 5 IMPROVING LLMS' ROBUSTNESS TO THE CHOICE OF DELIMITER

Having studied the brittleness of LLM on delimiters, this section explores approaches to improving LLM's robustness on delimiters: 1) Supervised Fine-Tuning (SFT) with randomly varying delimiters; and 2) Specifying the choice of delimiter in the prompt. Finally, this section provides a recommendation on the practical choice of delimiters for the best LLM performance.

**Supervised finetuning with randomly varying delimiters** We find naive supervised finetuning with randomly varying delimiters do not improve LLMs' sensitivity to delimiter choice, as shown in Appendix E. We suspect this stems from the distributional mismatch in SFT training data, which does not contain in-context examples.

**Specifying the choice of delimiter boosts typical performance** Next we explore whether specifying the delimiter choice in the prompt can boost LLMs' robustness to the choice of delimiter. We explicitly add a single line "*The following are multiple choice questions (with answers), separated by*

*X*" where X indicates the selected delimiter character. This removes the guess work for the language model to discern how each example is delineated. As shown in Table 2, specifying the delimiter choice improves model performance across choices of delimiters on all three benchmarks, ranging from 1.5% to 27.9%.

Table 2: **Specifying the delimiter choice in the prompt boosts average performance.** We show the average performance across different choices of delimiters when we include the choice of delimiter in the prompt. Specifically, we add an additional line before the demonstration examples stating examples are separated by delimiter X, where X corresponds to the delimiter used to separate examples.

| Benchmarks | Specify delimiter in prompt | Llama-3.1-70B-instruct | Llama-3.1-8B-instruct | Qwen2.5-7B-instruct |
|---|---|---|---|---|
| ARC-challenge | No | 68.7 | 44.2 | 48.1 |
| | Yes | **70.2** (+1.5) | **49.2** (+5.0) | **51.5** (+3.4) |
| commonsense-QA | No | 52.2 | 29.9 | 81.7 |
| | Yes | **77.8** (+25.6) | **57.8** (+27.9) | **83.1** (+1.5) |
| MMLU | Yes | 73.7 | 39.8 | 49.7 |
| | No | **76.5** (+2.9) | **42.7** (+14.2) | **63.9** (+2.8 ) |

**Practical delimiter recommendations**   In addition to the robustness strategy provided above, we find two delimiters "\n"and "!" performs well in the three families of models and three benchmarks. As shown in Table 3, we find the "\n"and "!" delimiters provide an average performance boost of 5.3% and 12.2% respectively, compared to the average performance across delimiters. More details are provided in Appendix Table 14.

Table 3: The performance gain for the best delimiter for each model family. The performance gain measures the average accuracy improvements across MMLU, ARC-challenge, commonsense-QA benchmarks.

| Models | Best delimiter | Performance gain |
|---|---|---|
| Llama-3.1-8B-instruct | \n | +12.21 |
| Qwen2.5-7B-instruct | ! | +5.26 |
| Gemma-2-9B-instruct | \n | +6.97 |

## 6 UNDERSTANDING HOW DELIMITERS STEER ATTENTION TO KEY TOKENS

To better understand how delimiters can affect LLMs behavior during inference, we consider the dictionary lookup task from Chen et al. (2024). To solve this task, the LLM must attend to the key token in the question and lookup its corresponding value in the context. We choose this in-context learning task specifically because we can precisely measure how the choice of delimiter steers the LLMs attention to the key token in the context, the necessary step to solve the task.

We find the choice of delimiter can heavily influence performance on the dictionary lookup task. For example, the performance of Llama-3.1-8B-instruct varies from 0% to 95% accuracy depending on the choice of delimiter as shown in the right panel of Figure 6.

To better understand how this choice of delimiter affects performance, we compare using a carriage return "\n" versus a space " " as delimiter for in-context examples. We then compute the attention scores for the lookup keys using the feature ablation method from Captum (Kokhlikyan et al., 2020). Specifically, for each prompt, we compare the attention scores for the **target key** tokens compared to the average attention scores of the tokens for the other **lookup keys** (see details in Appendix H). This isolates the effect of delimiters on whether the model is able to focus its attention weights on the relevant parts of the in-context samples to effectively solve the task. Attention scores for the target key with "\n" delimiter are boosted by 25% for Llama-3.1-8B-instruct, a statistically significant boost when using a paired t-test (t-statistic of 15.59). This suggests the right choice of delimiter improves performance by steering models' attention to focus on the relevant parts of the input.

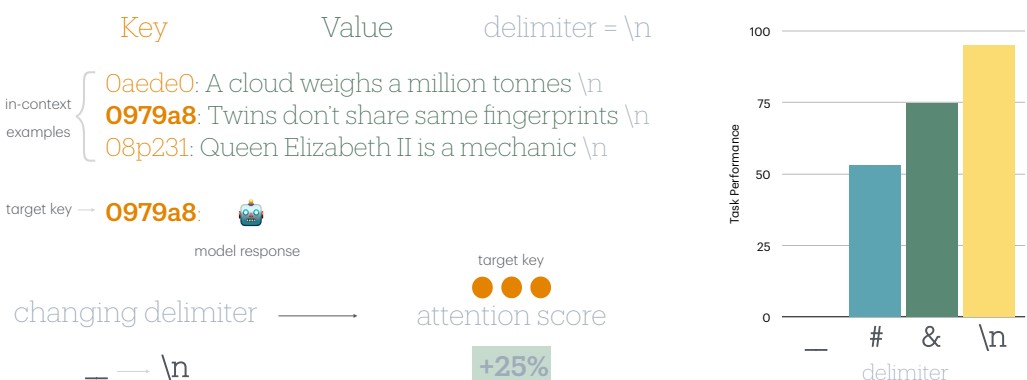

Figure 6: **Attention scores are steered towards the correct lookup key with the "\n"delimiter**. We measure a 25% statistically significant increase in the attention scores for the dictionary lookup task as we vary the delimiter. On the right panel, we show Llama-3.1-8B-instruct performance on the dictionary lookup task as we vary the choice of delimiter.

## 7 CONCLUSIONS

Our findings reveal the surprising importance of the single character used to separate examples in LLM evaluations. Far from an adversarial artifact, this choice reflects one of many options users face when interacting with chat models. We show how very reasonable delimiter selections from the set of ASCII characters can dramatically affect model performance across leading benchmarks and LLM families—with the important consequence that model rankings can be manipulated to put any model in the lead only by modifying the example delimiter. We propose two practical mitigations including a simple method of specifying the delimiter in the prompt, which we find consistently boosts performance across delimiters. We also provide practical recommendations for which delimiters lead on average to the best performance. Finally, we trace the effect of the delimiter in terms of how it steers attention. We find the right delimiter can significantly boost attention scores to focus on the relevant part of the input. In all, our findings suggest there is much more to learn about how the selection of delimiter interacts with queries, training data, and model architecture.

**Limitations and future work**    We only consider single character delimiter choices from the set of non-alphanumeric characters. While this provides a reasonably large set of straightforward options for separating examples, it's certainly possible to extend this set to characters outside of ASCII set or even consider multiple characters. However, even by only considering single character delimiters from ASCII, we find model performance can vary dramatically. In-context learning sample choices, how answers are delimited, and instructions via system prompt can also affect performance. We fix these choices and focus our study on the single character example delimiter in this work. Overall, more research is needed to better understand the complex interactions at inference and probe how such behavior emerges during large language model training.

## REPRODUCIBILITY

Our evaluations use the open-source eval-harness powering the Open-LLM-leaderboard available at `https://github.com/EleutherAI/lm-evaluation-harness`. We evaluate open-source models from three families Llama, Gemma, and Qwen all of which have checkpoints publicly available on HuggingFace. We use the same eval-harness codebase for evaluating GPT-4o using the API as documented in the repo's README. Our interpretability analysis uses the publicly available Captum PyTorch library using the feature ablation method (Kokhlikyan et al., 2020).

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

# APPENDIX

## A  LARGE LANGUAGE MODEL USAGE

We only use LLMs to polish writing.

## B  MMLU BENCHMARK SoTA EVOLUTION ACROSS YEARS

We attach the MMLU state-of-the-art performance evolution curve in Figure 7.

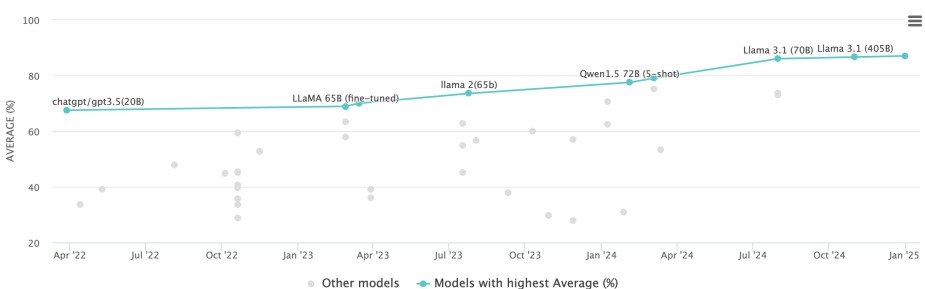

Figure 7: Evolution of MMLU state-of-the-art performance.

## C  THE SET OF DELIMITERS

We list all the considered ASCII delimiters in Table 4, which consist of all non-alphanumeric characters.

Table 4: The set of non-alphanumeric delimiters we consider in this paper.

| ! | # | $ | % | & | ( | ) | * | + | , |
|---|---|---|---|---|---|---|---|---|---|
| - | . | / | : | ; | < | = | > | ? | @ |
| [ | ] | ^ | _ | ' | { | \| | } | ~ | \n |

## D  ADDITIONAL EVALUATIONS

Table 5: MMLU summary statistics under different delimiters.

| Model | Highest | Lowest | Mean ± Std | Spread |
|---|---|---|---|---|
| Llama-3.1-8B-instruct | 51.99 (\n) | 33.68 (&) | 39.83 ± 3.73 | 18.31 |
| Gemma-2-9B-instruct | 60.16 (!) | 30.79 (?) | 43.36 ± 5.97 | 29.37 |
| Qwen2.5-7B-instruct | 65.02 (!) | 41.53 (#) | 49.73 ± 5.72 | 23.49 |
| Llama3.1-70B-instruct | 80.23 (!) | 64.18 ()) | 73.66 ± 4.08 | 16.05 |

We summarize the results for the models and common benchmarks considered in the main body of the paper in Table 5, Table 6, and Table 7. All instruction-tuned models consistently exhibit brittleness to the choice of delimiters.

## E  SFT USE VARYING DELIMITERS FAILED TO BOOST PERFORMANCE

In addition to prompting, we try to tune the LLM by using multiple different ASCII delimiters to replace the fixed "\n"in the chat template. We tune Llama-3.2-3B-instruct with the public Tulu

Table 6: ARC-challenge summary statistics under different delimiters.

| Model | Highest | Lowest | Mean ± Std | Spread |
|---|---|---|---|---|
| Llama3.1-8B-instruct | 47.70 (\n) | 41.89 (&) | 44.19 ± 1.18 | 5.81 |
| Gemma-2-9B-instruct | 69.71 (\n) | 59.22 (?) | 60.92 ± 1.75 | 10.49 |
| Qwen2.5-7B-instruct | 48.81 ($) | 47.01 (.) | 48.09 ± 0.44 | 1.80 |
| Llama3.1-70B-instruct | 69.80 (') | 67.24 (() | 68.66 ± 0.62 | 2.56 |

Table 7: commonsense-QA summary statistics under different delimiters.

| Model | Highest | Lowest | Mean ± Std | Spread |
|---|---|---|---|---|
| Llama3.1-8B-instruct | 51.35 (!) | 22.28 (?) | 29.89 ± 6.70 | 29.07 |
| Gemma-2-9B-instruct | 81.41 (\n) | 65.03 (?) | 79.95 ± 2.85 | 16.38 |
| Qwen2.5-7B-instruct | 82.64 (!) | 77.56 (_) | 81.68 ± 1.03 | 5.08 |
| Llama3.1-70B-instruct | 74.94 (!) | 34.97 (?) | 52.18 ± 9.77 | 39.97 |

SFT dataset (Lambert et al., 2024) using LoRA (Hu et al., 2022). We use the default settings from Axolotl maintainers and contributors (2023) with Lora r of 16 and Lora alpha of 32. MMLU results are included in Table 8 and 9. There is no consistent improvement as is the case with specifying the delimiter in the prompt. Similar results on ARC-challenge and commonsense-QA can be found in Table 10, 11 and Table 12, 13.

Table 8: MMLU, normal SFT.

| ! | # | $ | % | & | ( | ) | * | + | , |
|---|---|---|---|---|---|---|---|---|---|
| 57.77 | 57.29 | 57.34 | 57.56 | 57.43 | 57.20 | 57.00 | 57.93 | 57.64 | 57.29 |
| - | . | / | : | ; | < | = | > | ? | @ |
| 57.14 | 56.98 | 57.22 | 57.04 | 57.53 | 57.54 | 57.16 | 57.24 | 57.55 | 57.42 |
| [ | ] | ∧ | _ | ' | { | \| | } | ∼ | \n |
| 57.62 | 57.36 | 57.45 | 57.14 | 57.44 | 57.36 | 57.64 | 57.45 | 57.50 | 58.13 |

### E.1 MEASURING THE RELATIVE FREQUENCY OF DELIMITERS IN SFT DATA

In Table 15 we measure the relative frequency of delimiters in Tulu SFT, an open-source dataset used in instruction tuning of the Olmo-2 family of models. We find "\n" appears with the highest relative frequency.

## F DELIMITERS ARE NOT CONSISTENTLY BEST PER MODEL ACROSS BENCHMARKS

In this section, we provide the full results across ASCII delimiters with the datasets and models we have considered.

**Normal benchmarking.**

- MMLU. The results are included in Tables 16, 17, 18 and 19.
- commonsense-QA. The results are included in Tables 20, 21, 22 and 23.
- ARC-challenge. The results are included in Tables 24, 25, 26 and 27.

Table 9: MMLU, SFT with random delimiter choices.

| ! | # | $ | % | & | ( | ) | * | + | , |
|---|---|---|---|---|---|---|---|---|---|
| 56.00 | 55.54 | 55.61 | 55.26 | 55.36 | 54.44 | 54.42 | 55.53 | 55.68 | 54.88 |
| - | . | / | : | ; | < | = | > | ? | @ |
| 54.69 | 54.48 | 54.66 | 54.94 | 55.35 | 55.53 | 54.26 | 55.57 | 56.22 | 55.08 |
| [ | ] | ∧ | _ | ' | { | \| | } | ~ | \n |
| 55.06 | 55.56 | 55.38 | 54.65 | 55.00 | 55.02 | 55.89 | 56.02 | 55.88 | 56.50 |

Table 10: ARC-challenge, normal SFT.

| ! | # | $ | % | & | ( | ) | * | + | , |
|---|---|---|---|---|---|---|---|---|---|
| 45.05 | 44.37 | 44.28 | 44.11 | 43.69 | 44.28 | 44.54 | 44.11 | 44.20 | 43.17 |
| - | . | / | : | ; | < | = | > | ? | @ |
| 44.80 | 44.03 | 43.69 | 43.00 | 44.54 | 44.11 | 44.97 | 45.14 | 44.71 | 44.62 |
| [ | ] | ∧ | _ | ' | { | \| | } | ~ | \n |
| 44.88 | 44.97 | 44.20 | 44.28 | 44.54 | 44.28 | 45.05 | 44.62 | 44.37 | 45.14 |

**Benchmarking with delimiter specification.**  Note that Gemma-2-9B-instruct does not offer the system prompt choice, so we exclude this model from this set of evaluations.

- MMLU. The results are included in Tables 28, 29 and 30.
- commonsense-QA. The results are included in Tables 31, 32 and 33.
- ARC-challenge. The results are included in Tables 34, 35 and 36.

In Table 14 we show the best delimiter by model across topics in MMLU. We find relatively stable choices for the best delimiter for the smaller 7-9B parameter models and less stable choices that seem to depend on the topic for the larger 70B Llama model. This suggests in some cases interaction with the topic can also make a difference in the best choice of delimiter.

## G  EVALUATION ON CLOSED-SOURCE MODEL

In addition to the open-source model families we consider, we complement our result by using one representative closed-source model, GPT-4o, on the MMLU benchmark. Since the API does not offer log probability access, we have to switch to generation-based evaluation. In our preliminary experiment, we found the original filter implemented by Gao et al. (2024) failed to correctly parse the strings. Thus, we customize the filter (c.f. filter 1 and filter 2) and collect the summary statistics and full results in Table 1 and 37, respectively. The results are consistent with our observation on open-source models.

## H  MEASURING ATTENTION SCORES FOR THE DICTIONARY LOOKUP TASK

To measure the attention scores for the dictionary lookup task, we use the Captum library's (Kokhlikyan et al., 2020) feature ablation method for computing attention scores. We feed each input prompt and compare the attention scores of the target lookup key compared to the mean attention scores assigned to other lookup keys. We perform our analysis using Llama-3.1-8B-instruct to match the model used in other experiments. We then run a t-test on the attention scores of target versus lookup keys across inputs with the "\n"and space character delimiters from the dictionary lookup task. We find the attention scores for the target key do have a statistically significant difference across the delimiters with a t-statistic of 15.59 (p-value of $< 0.001$).

Table 11: ARC-challenge, SFT with random delimiter choices.

| ! | # | $ | % | & | ( | ) | * | + | , |
|---|---|---|---|---|---|---|---|---|---|
| 47.87 | 47.44 | 47.27 | 46.67 | 47.61 | 46.84 | 47.44 | 47.18 | 47.61 | 46.76 |
| - | . | / | : | ; | < | = | > | ? | @ |
| 47.44 | 46.33 | 47.35 | 47.01 | 47.78 | 47.35 | 47.27 | 47.53 | 47.10 | 47.27 |
| [ | ] | ∧ | _ | ' | { | \| | } | ∼ | \n |
| 47.10 | 47.53 | 47.53 | 46.50 | 46.93 | 46.50 | 47.53 | 46.93 | 47.53 | 47.10 |

Table 12: commonsense-QA, normal SFT.

| ! | # | $ | % | & | ( | ) | * | + | , |
|---|---|---|---|---|---|---|---|---|---|
| 73.87 | 74.20 | 74.28 | 74.04 | 73.96 | 73.79 | 73.55 | 74.28 | 73.63 | 73.14 |
| - | . | / | : | ; | < | = | > | ? | @ |
| 73.38 | 73.71 | 73.38 | 72.89 | 73.79 | 73.79 | 73.30 | 74.20 | 73.87 | 73.87 |
| [ | ] | ∧ | _ | ' | { | \| | } | ∼ | \n |
| 73.96 | 74.20 | 73.96 | 73.30 | 74.28 | 73.79 | 74.04 | 74.20 | 74.28 | 75.10 |

Filter for MMLU 1: Original filter used in Gao et al. (2024)

```
filter_list:
 name: get_response
 filter:
   # Filter everything after the first break line
   function: "regex"
   regex_pattern: "^(.*?)(?=\\n|$)"
   # Remove leading white spaces
   function: remove_whitespace
   # function to ignore right white spaces or line breaks
   function: "regex"
   regex_pattern: "^(.*?)\\s*$"
   function: take_first
```

Filter for MMLU 2: The filter we use to obtain the eval results

```
filter_list:
 - name: "custom-extract"
   filter:
    - function: "regex"
     regex_pattern: '(?i)(?:(?:the\s+)?(?:correct\s+)?(?:answer|choice|
         option|selection)\s*(?:is)?\s*:?|\A\s*)\(?\b([A-D])\b\)?(?:\.|\
         s|$)'
     # regex_pattern: 'answer is \(?([ABCDEFGHIJ])\)?'
     # regex_pattern: r".*[aA]nswer:\s*([A-J])",
    - function: "take_first"
```

Table 13: commonsense-QA, SFT with random delimiter choices.

| ! | # | $ | % | & | ( | ) | * | + | , |
|---|---|---|---|---|---|---|---|---|---|
| 73.22 | 73.05 | 73.55 | 73.05 | 72.56 | 72.73 | 72.73 | 73.14 | 72.65 | 72.32 |
| - | . | / | : | ; | < | = | > | ? | @ |
| 72.15 | 72.89 | 71.91 | 71.83 | 72.97 | 72.81 | 72.24 | 72.73 | 73.14 | 72.48 |
| [ | ] | ∧ | _ | ' | { | \| | } | ∼ | \n |
| 72.65 | 72.56 | 72.73 | 72.40 | 72.81 | 72.32 | 73.14 | 73.14 | 73.22 | 73.71 |

Table 14: Practical recommendations for best delimiter choice by topic.

| model | discipline | delimiter | accuracy |
|---|---|---|---|
| Llama-3.1-70B-instruct | STEM | $ | 71.6 |
| Llama-3.1-70B-instruct | humanities | ! | 84.3 |
| Llama-3.1-70B-instruct | other | ! | 79.3 |
| Llama-3.1-70B-instruct | social sciences | \n | 86.2 |
| Llama-3.1-8B-instruct | STEM | \n | 40.9 |
| Llama-3.1-8B-instruct | humanities | \n | 62.5 |
| Llama-3.1-8B-instruct | other | \n | 54.2 |
| Llama-3.1-8B-instruct | social sciences | \n | 61.3 |
| Qwen2.5-7B-instruct | STEM | ! | 59.1 |
| Qwen2.5-7B-instruct | humanities | ! | 66.1 |
| Qwen2.5-7B-instruct | other | ! | 63.4 |
| Qwen2.5-7B-instruct | social sciences | ! | 74.4 |
| Gemma-2-9B-instruct | STEM | ! | 47.5 |
| Gemma-2-9B-instruct | humanities | ! | 68.5 |
| Gemma-2-9B-instruct | other | ! | 61.5 |
| Gemma-2-9B-instruct | social sciences | ! | 62.9 |

Table 15: Relative frequency of delimiter in the Tulu SFT dataset. Specifically, tulu-3-sft-olmo-2-mixture used to train Olmo2.

| Delimiter | Relative Frequency |
|---|---|
| \n | 67.48% |
| ; | 0.82% |
| < | 7.81% |
| > | 7.93% |
| \| | 15.97% |

Table 16: MMLU, Llama-3.1-8B-instruct

| ! | # | $ | % | & | ( | ) | * | + | , |
|---|---|---|---|---|---|---|---|---|---|
| 47.98 | 38.84 | 43.38 | 36.79 | 33.68 | 37.17 | 40.56 | 41.75 | 37.50 | 41.17 |
| - | . | / | : | ; | < | = | > | ? | @ |
| 38.71 | 43.68 | 38.06 | 41.38 | 38.57 | 40.12 | 34.85 | 38.78 | 34.07 | 41.58 |
| [ | ] | ∧ | _ | ' | { | \| | } | ∼ | \n |
| 37.00 | 41.21 | 41.06 | 37.92 | 38.61 | 36.20 | 40.24 | 42.58 | 39.55 | 51.98 |

Table 17: MMLU, Qwen2.5-7B-instruct

| ! | # | $ | % | & | ( | ) | * | + | , |
|---|---|---|---|---|---|---|---|---|---|
| 65.02 | 41.53 | 40.05 | 50.48 | 41.85 | 57.96 | 51.91 | 56.18 | 53.74 | 54.76 |
| - | . | / | : | ; | < | = | > | ? | @ |
| 50.56 | 54.29 | 47.01 | 55.04 | 43.40 | 48.13 | 45.93 | 54.08 | 43.85 | 51.35 |
| [ | ] | ∧ | _ | ' | { | \| | } | ∼ | \n |
| 54.38 | 47.79 | 53.11 | 53.67 | 45.01 | 47.59 | 42.60 | 43.80 | 44.43 | 52.46 |

Table 18: MMLU, Gemma-2-9B-instruct

| ! | # | $ | % | & | ( | ) | * | + | , |
|---|---|---|---|---|---|---|---|---|---|
| 60.16 | 49.19 | 44.64 | 41.09 | 46.33 | 43.18 | 36.63 | 49.37 | 47.52 | 45.30 |
| - | . | / | : | ; | < | = | > | ? | @ |
| 38.58 | 43.41 | 41.01 | 36.40 | 48.17 | 45.21 | 34.75 | 36.85 | 30.79 | 49.15 |
| [ | ] | ∧ | _ | ' | { | \| | } | ∼ | \n |
| 47.38 | 38.65 | 43.42 | 42.54 | 42.06 | 46.52 | 41.69 | 36.65 | 40.02 | 54.02 |

Table 19: MMLU, Llama-3.1-70B-instruct

| ! | # | $ | % | & | ( | ) | * | + | , |
|---|---|---|---|---|---|---|---|---|---|
| 80.23 | 77.15 | 78.96 | 78.30 | 71.97 | 73.66 | 64.18 | 77.18 | 74.67 | 72.95 |
| - | . | / | : | ; | < | = | > | ? | @ |
| 66.86 | 72.90 | 68.57 | 69.56 | 76.51 | 75.27 | 69.80 | 71.68 | 66.32 | 75.13 |
| [ | ] | ∧ | _ | ' | { | \| | } | ∼ | \n |
| 75.92 | 71.10 | 77.12 | 70.71 | 76.76 | 71.18 | 75.34 | 74.23 | 75.40 | 80.07 |

Table 20: commonsense-QA, Llama-3.1-8B-instruct

| ! | # | $ | % | & | ( | ) | * | + | , |
|---|---|---|---|---|---|---|---|---|---|
| 51.35 | 25.23 | 33.17 | 24.82 | 26.62 | 26.62 | 30.38 | 31.94 | 25.39 | 27.44 |
| - | . | / | : | ; | < | = | > | ? | @ |
| 27.19 | 36.94 | 36.28 | 28.75 | 25.96 | 26.62 | 30.63 | 28.09 | 22.28 | 29.32 |
| [ | ] | ∧ | _ | ' | { | \| | } | ∼ | \n |
| 33.33 | 27.76 | 26.54 | 24.16 | 23.67 | 28.75 | 27.52 | 29.73 | 29.40 | 50.86 |

Table 21: commonsense-QA, Qwen2.5-7B-instruct

| ! | # | $ | % | & | ( | ) | * | + | , |
|---|---|---|---|---|---|---|---|---|---|
| 82.64 | 81.65 | 81.82 | 82.39 | 81.90 | 80.67 | 81.57 | 82.15 | 81.98 | 81.98 |
| - | . | / | : | ; | < | = | > | ? | @ |
| 79.52 | 81.08 | 81.41 | 82.39 | 82.39 | 82.31 | 80.67 | 81.82 | 82.47 | 82.23 |
| [ | ] | ∧ | _ | ' | { | \| | } | ∼ | \n |
| 82.23 | 82.23 | 81.98 | 77.56 | 82.15 | 82.56 | 81.16 | 81.65 | 81.49 | 82.39 |

Table 22: commonsense-QA, Gemma-2-9B-instruct

| ! | # | $ | % | & | ( | ) | * | + | , |
|---|---|---|---|---|---|---|---|---|---|
| 80.51 | 80.75 | 80.51 | 80.92 | 80.26 | 79.03 | 80.34 | 80.67 | 80.67 | 80.51 |
| - | . | / | : | ; | < | = | > | ? | @ |
| 79.85 | 80.10 | 80.34 | 80.59 | 80.26 | 80.59 | 80.10 | 80.75 | 65.03 | 80.84 |
| [ | ] | ∧ | _ | ' | { | \| | } | ∼ | \n |
| 80.59 | 80.43 | 80.59 | 80.59 | 80.51 | 80.51 | 80.59 | 80.26 | 80.59 | 81.41 |

Table 23: commonsense-QA, Llama-3.1-70B-instruct

| ! | # | $ | % | & | ( | ) | * | + | , |
|---|---|---|---|---|---|---|---|---|---|
| 74.94 | 55.28 | 54.71 | 49.71 | 45.70 | 54.55 | 36.12 | 66.01 | 49.63 | 49.06 |
| - | . | / | : | ; | < | = | > | ? | @ |
| 41.61 | 71.83 | 42.01 | 49.55 | 48.40 | 58.48 | 55.36 | 52.33 | 34.97 | 56.59 |
| [ | ] | ∧ | _ | ' | { | \| | } | ∼ | \n |
| 61.43 | 40.21 | 54.46 | 40.21 | 55.36 | 53.56 | 55.77 | 37.84 | 59.87 | 59.79 |

Table 24: ARC-challenge, Llama-3.1-8B-instruct

| ! | # | $ | % | & | ( | ) | * | + | , |
|---|---|---|---|---|---|---|---|---|---|
| 46.08 | 43.86 | 44.03 | 44.97 | 41.89 | 43.86 | 43.43 | 44.11 | 44.03 | 43.77 |
| - | . | / | : | ; | < | = | > | ? | @ |
| 43.77 | 45.14 | 42.41 | 44.28 | 44.54 | 43.09 | 43.86 | 43.86 | 43.09 | 44.88 |
| [ | ] | ∧ | _ | ' | { | \| | } | ∼ | \n |
| 43.43 | 43.94 | 44.97 | 44.45 | 44.71 | 42.92 | 43.09 | 46.84 | 44.80 | 47.70 |

Table 25: ARC-challenge, Qwen2.5-7B-instruct

| ! | # | $ | % | & | ( | ) | * | + | , |
|---|---|---|---|---|---|---|---|---|---|
| 47.61 | 48.12 | 48.81 | 48.21 | 48.63 | 47.95 | 48.04 | 48.72 | 48.38 | 47.95 |
| - | . | / | : | ; | < | = | > | ? | @ |
| 48.46 | 47.01 | 48.38 | 48.04 | 48.46 | 47.95 | 48.46 | 47.78 | 47.95 | 47.78 |
| [ | ] | ∧ | _ | ' | { | \| | } | ∼ | \n |
| 48.38 | 47.35 | 47.87 | 48.46 | 47.53 | 47.70 | 47.35 | 48.46 | 48.72 | 48.12 |

Table 26: ARC-challenge, Gemma-2-9B-instruct

| ! | # | $ | % | & | ( | ) | * | + | , |
|---|---|---|---|---|---|---|---|---|---|
| 61.77 | 60.84 | 60.49 | 61.09 | 59.64 | 60.07 | 60.49 | 60.75 | 59.98 | 60.92 |
| - | . | / | : | ; | < | = | > | ? | @ |
| 60.49 | 60.58 | 60.15 | 61.09 | 60.15 | 60.58 | 60.75 | 60.15 | 59.22 | 61.01 |
| [ | ] | ∧ | _ | ' | { | \| | } | ∼ | \n |
| 61.77 | 60.75 | 60.92 | 60.67 | 60.32 | 61.09 | 60.15 | 60.84 | 61.26 | 69.71 |

Table 27: ARC-challenge, Llama-3.1-70B-instruct

| ! | # | $ | % | & | ( | ) | * | + | , |
|---|---|---|---|---|---|---|---|---|---|
| 69.28 | 69.28 | 68.94 | 68.77 | 68.34 | 67.24 | 68.26 | 69.20 | 68.34 | 68.43 |
| - | . | / | : | ; | < | = | > | ? | @ |
| 68.26 | 68.94 | 67.92 | 68.86 | 68.26 | 69.03 | 68.09 | 69.11 | 67.75 | 68.86 |
| [ | ] | ∧ | _ | ' | { | \| | } | ∼ | \n |
| 67.49 | 68.26 | 69.03 | 68.86 | 69.80 | 68.34 | 69.37 | 68.52 | 69.54 | 69.54 |

Table 28: MMLU, Llama-3.1-8B-instruct + prompting

| ! | # | $ | % | & | ( | ) | * | + | , |
|---|---|---|---|---|---|---|---|---|---|
| 52.07 | 38.81 | 49.08 | 48.30 | 36.51 | 39.66 | 42.11 | 43.76 | 40.71 | 41.75 |
| - | . | / | : | ; | < | = | > | ? | @ |
| 37.16 | 46.96 | 40.17 | 44.42 | 40.12 | 42.45 | 34.57 | 41.87 | 42.39 | 40.91 |
| [ | ] | ∧ | _ | ' | { | \| | } | ∼ | \n |
| 40.29 | 45.50 | 45.59 | 44.77 | 42.95 | 37.00 | 39.35 | 48.00 | 42.28 | 49.96 |

Table 29: MMLU, Qwen2.5-7B-instruct + prompting

| ! | # | $ | % | & | ( | ) | * | + | , |
|---|---|---|---|---|---|---|---|---|---|
| 69.38 | 61.76 | 61.10 | 64.95 | 56.96 | 66.84 | 63.74 | 68.28 | 62.29 | 64.56 |
| - | . | / | : | ; | < | = | > | ? | @ |
| 61.47 | 64.15 | 65.90 | 64.75 | 62.26 | 65.67 | 66.63 | 69.27 | 66.54 | 63.14 |
| [ | ] | ∧ | _ | ' | { | \| | } | ∼ | \n |
| 64.55 | 65.67 | 64.53 | 67.92 | 60.69 | 67.83 | 53.68 | 68.11 | 57.14 | 57.95 |

Table 30: MMLU, Llama-3.1-70B-instruct + prompting

| ! | # | $ | % | & | ( | ) | * | + | , |
|---|---|---|---|---|---|---|---|---|---|
| 80.62 | 79.68 | 79.93 | 80.45 | 76.47 | 77.05 | 64.15 | 78.92 | 77.37 | 73.57 |
| - | . | / | : | ; | < | = | > | ? | @ |
| 68.15 | 74.01 | 71.75 | 68.86 | 78.53 | 78.23 | 73.01 | 74.71 | 76.60 | 78.40 |
| [ | ] | ∧ | _ | ' | { | \| | } | ∼ | \n |
| 79.26 | 77.34 | 79.46 | 75.94 | 79.21 | 78.34 | 77.77 | 78.80 | 78.23 | 80.43 |

Table 31: commonsense-QA, Llama-3.1-8B-instruct + prompting

| ! | # | $ | % | & | ( | ) | * | + | , |
|---|---|---|---|---|---|---|---|---|---|
| 72.40 | 50.45 | 73.14 | 74.37 | 41.61 | 36.69 | 65.03 | 65.03 | 53.40 | 45.13 |
| - | . | / | : | ; | < | = | > | ? | @ |
| 58.48 | 52.17 | 57.49 | 64.86 | 61.75 | 60.44 | 52.99 | 59.95 | 62.41 | 51.76 |
| [ | ] | ∧ | _ | ' | { | \| | } | ∼ | \n |
| 56.02 | 70.27 | 63.14 | 28.01 | 66.18 | 51.35 | 48.65 | 60.36 | 57.41 | 72.65 |

Table 32: commonsense-QA, Qwen2.5-7B-instruct + prompting

| ! | # | $ | % | & | ( | ) | * | + | , |
|---|---|---|---|---|---|---|---|---|---|
| 83.54 | 83.05 | 82.96 | 83.37 | 83.29 | 82.31 | 82.96 | 83.21 | 83.54 | 83.46 |
| - | . | / | : | ; | < | = | > | ? | @ |
| 81.98 | 81.65 | 82.88 | 83.70 | 83.54 | 82.88 | 83.13 | 83.29 | 83.78 | 82.47 |
| [ | ] | ∧ | _ | ' | { | \| | } | ∼ | \n |
| 83.21 | 83.54 | 83.46 | 82.56 | 83.78 | 82.96 | 83.05 | 83.78 | 83.62 | 82.88 |

Table 33: commonsense-QA, Llama-3.1-70B-instruct + prompting

| ! | # | $ | % | & | ( | ) | * | + | , |
|---|---|---|---|---|---|---|---|---|---|
| 82.06 | 80.59 | 81.00 | 80.51 | 81.24 | 77.15 | 64.21 | 81.49 | 80.75 | 73.55 |
| - | . | / | : | ; | < | = | > | ? | @ |
| 70.43 | 81.33 | 76.25 | 74.61 | 77.81 | 80.75 | 78.54 | 78.21 | 65.85 | 80.18 |
| [ | ] | ∧ | _ | ' | { | \| | } | ∼ | \n |
| 81.16 | 79.69 | 80.59 | 69.45 | 81.90 | 81.33 | 80.84 | 76.82 | 79.69 | 76.25 |

Table 34: ARC-challenge, Llama-3.1-8B-instruct + prompting

| ! | # | $ | % | & | ( | ) | * | + | , |
|---|---|---|---|---|---|---|---|---|---|
| 50.94 | 48.81 | 51.02 | 52.56 | 46.67 | 47.53 | 51.19 | 50.68 | 50.94 | 49.06 |
| - | . | / | : | ; | < | = | > | ? | @ |
| 46.16 | 49.06 | 46.59 | 51.19 | 49.57 | 48.12 | 48.63 | 48.21 | 48.55 | 50.09 |
| [ | ] | ∧ | _ | ' | { | \| | } | ∼ | \n |
| 46.25 | 49.49 | 51.62 | 46.76 | 47.87 | 49.66 | 47.53 | 51.19 | 50.17 | 50.00 |

Table 35: ARC-challenge, Qwen2.5-7B-instruct + prompting

| ! | # | $ | % | & | ( | ) | * | + | , |
|---|---|---|---|---|---|---|---|---|---|
| 50.26 | 48.63 | 50.85 | 50.43 | 50.26 | 54.61 | 53.41 | 49.57 | 51.02 | 54.27 |
| - | . | / | : | ; | < | = | > | ? | @ |
| 50.94 | 53.24 | 51.54 | 50.94 | 51.11 | 52.22 | 51.28 | 51.45 | 51.96 | 50.09 |
| [ | ] | ∧ | _ | ' | { | \| | } | ∼ | \n |
| 54.86 | 52.47 | 49.74 | 52.30 | 50.77 | 50.26 | 50.34 | 52.99 | 49.49 | 53.67 |

Table 36: ARC-challenge, Llama-3.1-70B-instruct + prompting

| ! | # | $ | % | & | ( | ) | * | + | , |
|---|---|---|---|---|---|---|---|---|---|
| 70.05 | 70.48 | 69.80 | 69.80 | 70.39 | 69.80 | 69.45 | 70.05 | 69.71 | 69.28 |
| - | . | / | : | ; | < | = | > | ? | @ |
| 70.73 | 70.22 | 70.65 | 70.39 | 70.14 | 70.56 | 70.48 | 70.99 | 70.22 | 70.14 |
| [ | ] | ∧ | _ | ' | { | \| | } | ∼ | \n |
| 69.97 | 70.22 | 69.97 | 70.48 | 69.71 | 69.97 | 70.56 | 70.65 | 70.22 | 70.05 |

Table 37: MMLU (generation), GPT-4o

| ! | # | $ | % | & | ( | ) | * | + | , |
|---|---|---|---|---|---|---|---|---|---|
| 61.00 | 77.55 | 73.36 | 78.44 | 69.43 | 68.32 | 68.69 | 72.58 | 72.50 | 69.31 |
| - | . | / | : | ; | < | = | > | ? | @ |
| 69.34 | 72.17 | 68.00 | 48.89 | 69.94 | 76.91 | 65.69 | 72.33 | 32.97 | 78.59 |
| [ | ] | ∧ | _ | ' | { | \| | } | ∼ | \n |
| 69.75 | 65.07 | 75.35 | 70.81 | 72.68 | 74.25 | 75.92 | 69.76 | 72.30 | 67.14 |

