# OpenReview forum: "A Single Character Can Make or Break your LLM Evals"
_ICLR.cc/2026/Conference — Submitted to ICLR 2026_

### Official Review · Reviewer_96SC · 2025-10-28

**Soundness:** 2
**Presentation:** 2
**Contribution:** 1
**Rating:** 2
**Confidence:** 3

**Summary:**

This paper shows that a single character used to delimit in-context examples can drastically change LLM benchmark outcomes. Varying only the delimiter across 30 ASCII symbols yields swings up to ±30 points on MMLU, with rankings reversible across Llama, Qwen, and Gemma; GPT-4o shows even larger spread. The brittleness persists across topics, model scales (8B→70B), and few-shot counts, and extends to non-MCQ ICL tasks (Banking77, TACRED). Attention analyses on a dictionary-lookup task suggest good delimiters steer attention to key tokens. Naive SFT with random delimiters fails to help, but explicitly stating the delimiter in the prompt improves robustness.

**Strengths:**

1. The paper isolates a single, realistic prompt variable (example delimiter) and demonstrates large, reproducible accuracy swings and rank reversals across multiple model families, sizes, tasks, and even a closed model. This challenges current evaluation practices and has immediate implications for leaderboard validity and prompt design.

2. Beyond documenting brittleness, the work offers actionable fixes (explicitly specifying the delimiter; robust defaults like “\n”/“!”) and provides an interpretability angle showing how good delimiters steer attention to task-relevant tokens, strengthening the causal story rather than remaining a purely empirical observation.

**Weaknesses:**

1. A major concern is that the paper evaluates instruction-tuned models on multiple-choice via next-token probabilities and few-shot ICL, whereas in practice instruction-tuned models are now commonly evaluated in zero-shot generation mode with output parsing rather than prob-based scoring; this mismatch raises serious questions about the real-world relevance of the conclusions. It is necessary to see what tokens indeed will models generate when we switch to different delimiters instead of using next token probablity as proxy.

2. The proposed mitigations are narrow and may not generalize: SFT with random delimiters fails, and “state the delimiter” or “prefer \n/!” could be template-, tokenizer-, or dataset-dependent; the paper also does not study multi-character or structured delimiters (e.g., XML/Markdown blocks) or production normalization pipelines, limiting the breadth of the claims.

**Questions:**

See Weaknesses

---

### Official Review · Reviewer_AJKt · 2025-10-31

**Soundness:** 3
**Presentation:** 3
**Contribution:** 2
**Rating:** 4
**Confidence:** 3

**Summary:**

This paper studies how LLM evaluations are affected by the formatting of few-shot examples. The authors focus on a specific variable: the single character used to separate in-context examples, termed the "example delimiter". They claim this small choice can greatly change model response quality and cause large performance shifts on standard benchmarks. The paper also looks into why this happens by checking attention scores and suggests practical ways to make models more stable, like specifying the delimiter in the prompt.

The experiments test 30 different non-alphanumeric ASCII characters as delimiters across model families (Llama, Qwen, Gemma) and popular benchmarks (MMLU, ARC-challenge, commonsense-QA). The results show that MMLU performance can change by 23% based on the delimiter choice. Specific models showed performance gaps from 18.3% to 29.4%. This effect appears across different topics and is not solved by using larger models (eg, 8B vs 70B). The study also finds that telling the model the delimiter in the prompt can raise average performance, and it points to \n and ! as generally effective choices.

**Strengths:**

1. The paper's primary strength is its clear finding that a single-character delimiter can significantly impact model performance. This observation effectively challenges the robustness and reliability of many current LLM evaluation protocols.

2. The authors support this claim with a comprehensive set of experiments. The study is thorough, covering multiple leading model families and several mainstream benchmarks.

3. The paper investigates mitigation strategies, including both SFT and a prompting-based fix. This is supplemented by an initial mechanistic analysis using attention probing.

**Weaknesses:**

**Major Issues:**

My primary concern is that the experimental setup feels unnatural and may not reflect real-world usage. The study appears to function more as an artificial stress test than a practical evaluation of common scenarios.

First, many of the 30 tested ASCII characters (e.g., '?', '(', ']') are not logical or common choices for separating examples. These characters might confuse the model or be misinterpreted. For instance, a prompt ending in "Answer: A?" could be interpreted as questioning the demonstration itself. In such cases, a significant drop in performance is not surprising.

Second, real-world prompts and evaluation harnesses often use more robust, multi-character separators for clarity (e.g., \n\n, \n---\n, or tags like <end_of_example>). The paper's focus on a single character feels like a limited scenario.

Given these points, the significance of the findings is somewhat limited. The paper would be much stronger if it focused on performance variations between commonly used, reasonable delimiters (both single and multi-character) rather than a wide set of mostly illogical ones.

Furthermore, the novelty of the core finding is also a concern. It is well-established (e.g., Sclar et al., 2024) that LLMs are extremely sensitive to prompt formatting. Unless the authors can provide evidence that this specific single-character delimiter variance is a widespread and overlooked problem in practice, the paper's contribution feels more like an incremental (though extreme) example of this known issue, rather than a new, fundamental insight.

**Minor Issues:**

1. The analysis of the phenomenon is somewhat shallow. For instance, if one chooses to use an unconventional delimiter like '?', then specifying this choice in the prompt (the paper's proposed solution) seems less like a novel mitigation strategy and more like a necessary part of a well-formed prompt.

2. The attention probing analysis explains that attention is steered, but it doesn't sufficiently explore why a character like \n is more effective at steering attention than a character like &. This likely relates to data distributions, which is not deeply investigated.

**Questions:**

See the above weaknesses.

---

### Official Review · Reviewer_hF5U · 2025-11-04

**Soundness:** 3
**Presentation:** 2
**Contribution:** 1
**Rating:** 2
**Confidence:** 4

**Summary:**

This paper investigates the understudied impact of example delimiters on large language model (LLM) evaluations. It systematically tests 30 non-alphanumeric ASCII delimiters (e.g., \n, !, &) across open-source LLMs (Llama, Qwen, Gemma) and closed-source GPT-4o, using benchmarks like MMLU, ARC-challenge, and commonsense-QA.
Key findings include: 1) A single delimiter change causes up to 29.4% performance fluctuation on MMLU (equivalent to 3 years of LLM progress since 2022) and enables manipulating model rankings. 2) This brittleness is pervasive—present across models, benchmarks, topics, and unmitigated by scaling (e.g., Llama-3.1 70B shows larger fluctuation than 8B). 3) Mechanistically, effective delimiters (e.g., \n) boost attention to key input tokens by 25% (statistically significant).
The paper proposes practical solutions: explicitly specifying delimiters in prompts improves robustness (e.g., +14.2% MMLU for Qwen2.5-7B-instruct) and recommends \n and ! as optimal delimiters. It highlights critical flaws in current LLM evaluation protocols and provides actionable insights for reliable LLM assessment.

**Strengths:**

It is an interesting result that a single character can lead to such a significant drop in in-context learning (ICL) performance. This paper validates this observation across multiple standard benchmarks and a range of models.

**Weaknesses:**

Significant issue:
1. The finding is somewhat obvious. The sensitivity of prompts is a widely studied and well-known problem for LLMs.
2. Lack of in-depth analysis. The paper mainly discusses how large the gap is with different delimiters. However, it fails to clearly point out either how to resolve this problem or what the root cause of this problem is. Is it the frequency of the delimiters in the pre-training data? How do your findings help understand LLM's mechanism?
3. The research question seems insignificant. If we all use \n, the issue is no longer valid?

Minor issue:
1. Presentation issues: Legends in figures (i.e., Figures 1-3) are too small to recognize. Also, you only present the max and the min values. You should also show the average and variance of scores with different delimiters. It would help us understand how severe the problem is.

**Questions:**

NA

---

### Author Response · Authors · 2025-11-18

We appreciate reviewers’ effort in engaging with our work. We fear, however, reviewers have not understood the primary contribution of our work and would like a chance to clarify below.

- **Soundness of our setup**: our evaluations use the standard protocol from the Eval Harness repo (with >10k stars) powering the HuggingFace LM Leaderboard. We report performance on benchmarks such as MMLU and ARC, commonly used to measure the capabilities of leading models—appearing in DeepSeek, Llama, Qwen, Gemini and nearly every other major model release. We report numbers on both probability-based evaluations as well as generation based (see GPT-4o results), across a range of leading LLMs, which confirm the generality of our findings across model families, sizes, and protocols. We find, even very reasonable choices of delimiters, such as “#”, a widely used character to designate sections in markdown, can lead to a considerable fluctuation in performance.

- **“Finding is obvious”**: reviewers characterized the large fluctuation in LLM performance as “well-established” and “obvious”. Today the AI community, including researchers, rely on the very benchmarks we evaluated, MMLU, ARC, etc. to rank model performance. The implication of our work undermines the validity of this pervasive practice. We show LLM ranking can be manipulated only by modifying the single character separating examples, thus can not be trusted.

We hope these clarifications reaffirm the soundness of our methodology and the pressing importance of our findings to reconsider the basis for LLM rankings.

---

### Meta-Review · Area_Chair_PRFo · 2026-01-11

**Summary:**

The paper studies a specific prompt-formatting choice in few-shot evaluation: the *single-character delimiter* separating in-context examples. By sweeping 30 non-alphanumeric ASCII delimiters across multiple model families (Llama, Qwen, Gemma) and GPT-4o, and across benchmarks (e.g., MMLU, ARC-Challenge, CommonsenseQA; also some non-MCQ ICL tasks), the authors report large performance swings (up to ~29.4 points on MMLU) and show that model rankings can be reversed by delimiter choice. The paper further reports that this brittleness persists across topics and model scales (8B→70B). For mechanism, an attention-based probe suggests “good” delimiters correlate with increased attention to key tokens. For mitigation, the paper reports that explicitly stating the delimiter in the prompt improves robustness; naive SFT with randomized delimiters does not.

### Strengths
- **Clear empirical phenomenon with broad coverage.** Multiple reviewers (AJKt, 96SC, hF5U) agree the paper demonstrates large, reproducible swings across several model families and standard benchmarks.
- **Immediate implication for benchmark/leaderboard validity.** The demonstrated “rank reversals” directly challenge the stability of common evaluation setups used in practice (as also emphasized in the author comment referencing Eval Harness usage).

### Weaknesses
- **Concerns about real-world relevance of the exact evaluation protocol.** Reviewer 96SC raises a soundness/relevance concern that prob-based MCQ evaluation of instruction-tuned models may mismatch common current practice (zero-shot generation + parsing), and asks to verify brittleness under generation-based evaluation rather than only next-token probability scoring.
- **Limited depth on root cause / why certain delimiters work.** Reviewers (hF5U, AJKt) find the mechanistic explanation insufficiently developed (e.g., whether effects relate to data frequency/tokenization; why `\n` works better than other symbols).
- **Novelty framing.** Two reviewers (hF5U, AJKt) view “prompt sensitivity” as known and want stronger evidence that *this specific delimiter issue* is a practically overlooked failure mode rather than an expected instance of prompt brittleness.

My decision hinges on whether the paper’s evidence supports its broad implications for evaluation reliability *under commonly used evaluation protocols*. While the paper demonstrates a striking sensitivity under its chosen setup, a key reviewer concern (96SC) questions whether the main results carry over to generation-based evaluation (a core “real-world relevance” issue). Additionally, concerns that the delimiter set is dominated by unrealistic choices and that multi-character/structured separators are not evaluated (AJKt) limit how strongly we can generalize the headline claim. These are not mere presentation issues; they directly affect the scope and soundness of the paper’s conclusions about benchmark validity.

**Reviewer Concerns:**

**Addressed:**
- **Protocol justification / relevance to existing leaderboards.** The rebuttal provides a reasonable justification that the setup aligns with widely used harnesses, which partially addresses “unrealistic setup” concerns.

**Still outstanding:**
- **Generation-based evaluation coverage for the main claims (esp. open models / instruction-tuned models).** Reviewer 96SC’s request is specifically about switching away from next-token probability scoring as the proxy and checking what models *generate* under delimiter changes. The rebuttal mentions generation-based results for GPT-4o, but does not clearly resolve whether the central brittleness/rank-reversal claims hold broadly under generation-based evaluation across the studied open models/benchmarks.
- **Scope concerns about delimiter choice realism and multi-character delimiters.** The rebuttal states that even reasonable delimiters can cause fluctuations, but does not address the broader limitation that many separators tested may be uncommon and that multi-character/structured separators are not studied (AJKt, 96SC).
- **Root-cause depth.** The rebuttal does not materially expand the mechanistic explanation beyond what reviewers considered shallow.

**Reviewer Scores:**

No changes expected.

---

### Decision · Program_Chairs · 2026-01-26

Reject